# ENSEMBLES OF GENERATIVE ADVERSARIAL NETWORKS FOR DISCONNECTED DATA

## ABSTRACT

Most computer vision datasets are composed of disconnected sets, such as images of different objects. We prove that distributions of this type of data cannot be represented with a continuous generative network without error, independent of the learning algorithm used. Disconnected datasets can be represented in two ways: with an ensemble of networks or with a single network using a truncated latent space. We show that ensembles are more desirable than truncated distributions for several theoretical and computational reasons. We construct a regularized optimization problem that rigorously establishes the relationships between a single continuous GAN, an ensemble of GANs, conditional GANs, and Gaussian Mixture GANs. The regularization can be computed efficiently, and we show empirically that our framework has a performance sweet spot that can be found via hyperparameter tuning. The ensemble framework provides better performance than a single continuous GAN or cGAN while maintaining fewer total parameters.

## 1 INTRODUCTION

Generative networks, such as generative adversarial networks (GANs) (Goodfellow et al., 2014) and variational autoencoders (Kingma & Welling, 2013), have shown impressive performance in generating highly realistic images that were not observed in the training set (Karras et al., 2017; 2019a;b). However, even state of the art generative networks such as BigGAN (Brock et al., 2018) generate poor quality imagery if conditioned on certain classes of ILSVRC2012 (Russakovsky et al., 2015). We argue that this is due to the inherent *disconnected structure* of the data.

In this paper, we theoretically analyze the effects of disconnected data on GAN performance. By disconnected, we mean that the data points are drawn from an underlying topological space that is disconnected (the rigorous definition is provided below in Section 3.1). As an intuitive example, consider the collection of all images of badgers and all images of zebras. These two sets are disconnected, because images of badgers do not resemble images of zebras, and modeling the space connecting these sets does not represent real images of animals.

We rigorously prove that one cannot use a single continuous generative network to learn a data distribution perfectly under the disconnected data model. Because generative networks are continuous, they cannot map a connected latent space ($\mathbb{R}^{\ell}$) into the disconnected image space, resulting in the generation of data outside of the true data space. In related work, (Khayatkhoei et al., 2018) has empirically studied disconnected data but does not formally prove the results in this paper. In addition, the authors use a completely unsupervised approach to attempt to find the disconnected components as a part of learning. In contrast, we use class labels and hence work in the supervised learning regime.

Our suggested approach to best deal with disconnected data is to use ensembles of GANs. We study GANs in particular for concreteness and because of their widespread application; however, our methods can be extended to other generative networks with some modification. Ensembles of GANs are not new, e.g., see (Nguyen et al., 2017; Ghosh et al., 2018; Tolstikhin et al., 2017; Arora et al., 2017), but there has been limited theoretical study of their properties. We prove that ensembles can learn the data distribution under the disconnected data assumption and study their relationship to single GANs. Specifically, we develop a first-of-its-kind theoretic framework that relates single GANs, ensembles of GANs, conditional GANs, and Gaussian mixture GANs. The framework makes

it easy to, e.g., develop regularized GAN ensembles that encourage parameter sharing, which we show outperform cGANs and single GANs.

While our primary focus here is on theoretical insight, we also conduct a range of experiments to demonstrate empirically that the performance (measured in terms of FID (Heusel et al., 2017), MSE to the training set (Metz et al., 2016), Precision, and Recall (Sajjadi et al., 2018)) increases when we use an ensemble of WGANs over a single WGAN on the CIFAR-10 dataset (Krizhevsky & Hinton, 2009). The performance increase can be explained in terms of three contributing factors: 1) the ensemble has more parameters and hence has higher capacity to learn complex distributions, 2) the ensemble better captures the disconnected structure of the data, and 3) parameter sharing among ensemble networks enables successful joint learning, which we observe can increase performance.

We summarize our contributions as follows:

- We prove that generative networks, which are continuous functions, cannot learn the data distribution if the data is disconnected (Section 3.2). The disconnected data model is defined in Section 3.1, where we argue that it is satisfied in many common datasets, such as MNIST, CIFAR-10, and ILSVRC2012. Restricting the generator to a disconnected subset of the domain is one solution (Section 3.3), but we study a better solution: using ensembles.

- We demonstrate how single GANs and ensembles are related (Section 4.1). We then prove that ensembles are able to learn the true data distribution under our disconnected data model (Section 4.2). Finally, we demonstrate that there is an equivalence between ensembles of GANs and common architectures such as cGANs and GM-GANs due to parameter sharing between ensemble components (Section 4.3).

- We empirically show that, in general, an ensemble of GANs outperforms a single GAN (Section 5.1). This is true even if we reduce the number of parameters used in an ensemble so that it has fewer total parameters than a single GAN (Section 5.2). Finally, we empirically show that parameter sharing among ensemble networks leads to better performance than a single GAN (Section 5.3) or even a cGAN (Section 5.4).

## 2    BACKGROUND AND RELATED WORK

### 2.1    GENERATIVE ADVERSARIAL NETWORKS (GANS)

GANs are generative neural networks that use an adversarial loss, typically from another neural network (Goodfellow et al., 2014). In other words, a GAN consists of two neural networks that compete against each other. The generator $G : \mathbb{R}^{\ell} \to \mathbb{R}^{p}$ is a neural network that generates $p$-dimensional images from an $\ell$-dimensional latent space. The discriminator $D : \mathbb{R}^{p} \to (0, 1)$ is a neural network which is trained to classify between the training set and generated images. As compositions of continuous functions (Goodfellow et al., 2016), both $G$ and $D$ are continuous.

$G$ has parameters $\boldsymbol{\theta}_G \in \mathbb{R}^{|\boldsymbol{\theta}_G|}$, where $|\boldsymbol{\theta}_G|$ is the possibly infinite cardinality of $\boldsymbol{\theta}_G$. Similarly, $D$ has parameters $\boldsymbol{\theta}_D \in \mathbb{R}^{|\boldsymbol{\theta}_D|}$. The latent, generated, and data distributions are $P_{\boldsymbol{z}}, P_G$, and $P_{\mathcal{X}}$, respectively. We train this network by solving the following optimization problem:

$$\min_{\boldsymbol{\theta}_G} \max_{\boldsymbol{\theta}_D} V(\boldsymbol{\theta}_G, \boldsymbol{\theta}_D) = \min_{\boldsymbol{\theta}_G} \max_{\boldsymbol{\theta}_D} \mathbb{E}_{\boldsymbol{x} \sim P_{\mathcal{X}}}[\log D(\boldsymbol{x})] + \mathbb{E}_{\boldsymbol{z} \sim P_{\boldsymbol{z}}}[\log(1 - D(G(\boldsymbol{z})))]. \tag{1}$$

Here we write $\min$ and $\max$ instead of $\mathrm{minimize}$ and $\mathrm{maximize}$ for notational compactness, but we are referring to an optimization problem. The objective of this optimization is to learn the true data distribution, i.e., $P_G = P_{\mathcal{X}}$. Alternatively, we can use the Wasserstein distance instead of the typical cross-entropy loss: $V(\boldsymbol{\theta}_G, \boldsymbol{\theta}_D) = \mathbb{E}_{\boldsymbol{x} \sim P_{\mathcal{X}}} D(\boldsymbol{x}) - \mathbb{E}_{\boldsymbol{z} \sim P_{\boldsymbol{z}}} D(G(\boldsymbol{z}))$ restricted to those $\boldsymbol{\theta}_G, \boldsymbol{\theta}_D$ which force $D$ to be 1-Lipschitz as done in the WGAN paper (Arjovsky et al., 2017). Thus, we will use $V$ to denote either of these two objective functions.

### 2.2    GANS THAT TREAT SUBSETS OF DATA DIFFERENTLY

**Ensembles of GANs.**    Datasets with many different classes, such as ILSVRC2012 (Russakovsky et al., 2015), are harder to learn in part because the relationship between classes is difficult to quantify. Some models, such as AC-GANs (Odena et al., 2017), tackle this complexity by training different

models on different classes of data in a supervised fashion. In the AC-GAN paper, the authors train 100 GANs on the 1000 classes of ILSVRC2012. The need for these ensembles is not theoretically studied or justified beyond their intuitive usefulness.

Several ensembles of GANs have been studied in the unsupervised setting, where the modes or disconnected subsets of the latent space are typically learned (Pandeva & Schubert, 2019; Hoang et al., 2018; Khayatkhoei et al., 2018) with some information theoretic regularization as done in (Chen et al., 2016). These are unsupervised approaches which we do not study in this paper. Models such as SGAN (Chavdarova & Fleuret, 2018) and standard GAN ensembles (Wang et al., 2016) use several GANs in part to increase the capacity or expressibility of GANs. Other ensembles, such as Dropout-GAN (Mordido et al., 2018), help increase robustness of the generative network.

**Conditional GANs (cGANs).**  Conditional GANs (Mirza & Osindero, 2014) attempt to solve the optimization problem in (1) by conditioning on the class $y$, a one-hot vector. The generator and discriminator both take $y$ as an additional input. This conditioning can be implemented by having the latent variable be part of the input, e.g., the input to the generator will be $[z^T \ y^T]^T$ instead of just $z$.

Typically, conventional cGANs have the following architecture modification. The first layer has an additive bias that depends on the class vector $y$ and the rest is the same. For example, consider a multilayer perceptron, with matrix $W$ in the first layer. Converting this network to be conditional would result in the following modification to the matrix in the first layer:

$$W_{\text{conditional}} \begin{bmatrix} x \\ y \end{bmatrix} = \begin{bmatrix} W & B \end{bmatrix} \begin{bmatrix} x \\ y \end{bmatrix} = Wx + By = Wx + B_{\cdot, k}.$$

Hence, we can think of $B$ as a matrix with columns $B_{\cdot, k}, k \in \{1, \ldots, K\}$ being bias vectors and $W$ being the same as before. We pick a bias vector $B_{\cdot, k}$ based on what class we are conditioning on but the other parameters of the network are held the same, independent of $k$. This is done to both the generator and the discriminator. Some cGANs condition on multiple layers, such as BigGAN (Brock et al., 2018), or on different types of layers, such as convolutional layers, but our formulation here extends clearly to those other architectures.

**Gaussian Mixture GANs (GM-GANs).**  The latent distribution $P_z$ is typically chosen to be either uniform, isotropic Gaussian, or truncated isotropic Gaussian (Goodfellow et al., 2014; Radford et al., 2015; Brock et al., 2018). We are not restricted to these distributions; research has been conducted in extending and studying the affect of using different distributions, such as a mixture of Gaussians (Ben-Yosef & Weinshall, 2018; Gurumurthy et al., 2017).

## 3 Continuous generative networks cannot model distributions drawn from disconnected data

### 3.1 Disconnected data model

We begin by introducing a new data model that accounts for disconnected data. Typical datasets with class labels satisfy this model; we provide additional examples below.

**Definition 1** (**Disconnected data model**). We assume that the data lies on $K$ disjoint, compact sets $\mathcal{X}_k \subset \mathbb{R}^p, k \in \{1, \ldots, K\}$ so that the whole data lies on the disjoint union of each component: $\bigsqcup_{k=1}^{K} \mathcal{X}_k = \mathcal{X}$. Moreover, we assume that each component $\mathcal{X}_k$ is connected (Rudin, 1964). We then draw data points from these sets in order to construct our finite datasets.

In Definition 1, we let each $\mathcal{X}_k$ be compact in order to remove the degenerate case of having two components $\mathcal{X}_k$ and $\mathcal{X}_j$ that are arbitrarily close to one another, which is possible if we only assume that $\mathcal{X}$ is closed and disjoint. If that is the case, there are trivial counter-examples (see the appendix) to the theorems proved below.

**Lemma 1.** *$\mathcal{X}$ is a disconnected set, and $\mathcal{X}_j$ is disconnected from $\mathcal{X}_k$ for $j \neq k$.*

Disconnected datasets are ubiquitous in machine learning (Khayatkhoei et al., 2018; Hoang et al., 2018; Pandeva & Schubert, 2019). For example, datasets with discrete labels (typical in classification problems) will often be disconnected. We study this disconnected data property, because generative networks are unable to learn the distribution supported on such a dataset, as we show below.

### 3.2 CONTINUOUS GENERATIVE NETWORKS CANNOT REPRESENT A DISCONNECTED DATA DISTRIBUTION EXACTLY

In this section, we prove that, under Definition 1, continuous generative networks cannot learn the true data distribution exactly due to model misspecification.

Suppose that $(\Omega, \mathcal{F}, P_{\boldsymbol{z}})$ is a probability space with $P_{\boldsymbol{z}}$ being the distribution of the random vector $\boldsymbol{z} : \Omega \to \mathbb{R}^{\ell}$. We assume that $P_{\boldsymbol{z}}$ is equivalent to the Lebesgue measure $\lambda$. This just means that $\lambda(\boldsymbol{z}(A)) = 0$ if and only if $P_{\boldsymbol{z}}(A) = 0$ for any set $A \in \mathcal{F}$. This is true for a Gaussian distribution, for example, which is commonly used as a latent distribution in GANs (Arjovsky et al., 2017). The transformed (via the generative network $G$) random vector $\boldsymbol{x} = G \circ \boldsymbol{z} : \Omega \to \mathbb{R}^p$ is determined by the original probability measure $P_{\boldsymbol{z}}$ but is defined on the induced probability space $(\Omega', \mathcal{F}', P_G)$.

**Theorem 1.** *If $G$ can generate from multiple components of $\mathcal{X}$ (say $\mathcal{X}_1$ and $\mathcal{X}_2$), then the probability of generating samples outside of $\mathcal{X}$ is positive: $P_G(\boldsymbol{x} \in \mathbb{R}^p \backslash \mathcal{X}) > 0$. Otherwise if we can only generate from one component (say $\mathcal{X}_1$), then $P_G(\mathcal{X}_i) = 0$ for $x \in \{2, \ldots, K\}$.*

The continuity of $G$ is the fundamental reason why Theorem 1 is true. A continuous function cannot map a connected space to a disconnected space. This means that all generative networks must generate samples outside of the dataset if the data satisfies Definition 1.

Suppose that our data is generated from the true random vector $\boldsymbol{x}_{\text{data}} : \Omega' \to \mathbb{R}^p$ using the probability distribution $P_{\mathcal{X}}$. Also, suppose that we learn $P_G$ by training a generative network.

**Corollary 1.** *Under Definition 1, we have that $d(P_G, P_{\mathcal{X}}) > 0$ for any distance metric $d$ and any learned distribution $P_G$.*

From Corollary 1, we see that learning the data distribution will incur irreducible error under Definition 1 because our data model and the model that we are trying to train do not match. Hence, we need to change which models we consider when we train in order to best reflect the structure of our data. At first thought a discontinuous $G$ might be considered, but that would require training $G$ without backpropagation. Instead, we focus on restricting $G$ to a discontinuous domain (Section 3.3) and training an ensemble of GANs (Section 4.1) as two possible solutions.

### 3.3 RESTRICTING THE GENERATOR TO A DISCONNECTED SUBSET OF THE LATENT DISTRIBUTION

In this section, we study how we can remove the irreducible error in Theorem 1 from our models after training. Suppose that we train a generator $G$ on some data so that $G(\mathbb{R}^{\ell}) \supset \mathcal{X}$. Therefore, we can actually generate points from the true data distribution. We know that the distributions cannot be equal because of Theorem 1, implying that if we restrict the domain of $G$ to the set $Z = G^{-1}(\mathcal{X})$ then $G(Z) = \mathcal{X}$. The next theorem shows how the latent distribution is related to restricting the domain of $G$.

**Theorem 2 (Truncating the latent space reduces error).** *Suppose that $P_{\boldsymbol{z}}(Z) > 0$ and let the generator $G$ learn a proportionally correct distribution over $\mathcal{X}$. In other words, there exists a real number $c \in \mathbb{R}$ so that*

$$P_G(A) = cP_{\mathcal{X}}(A) \qquad A \in \mathcal{F}', A \subset \mathcal{X}.$$

*Then, we use the truncated latent distribution defined by $P_{\boldsymbol{z}_T}(B) = 0$ for all $B \in \mathcal{F}$ that satisfy $B \cap Z = \varnothing$. This allows us to learn the data distribution exactly, i.e.*

$$P_{G|_Z}(A) = P_{\mathcal{X}}(A) \qquad A \in \mathcal{F}'.$$

We write $P_{G|_Z}$ because, by truncating the latent distribution, we effectively restrict $G$ to the domain $Z$. Theorem 2 shows that if we learn the data distribution approximately by learning a proportional distribution, then we can learn the true data distribution by truncating our latent distribution. By 4.22 in (Rudin, 1964), $Z$ must be disconnected, which implies that a disconnected latent distribution is a solution to remove the irreducible error in Theorem 1.

Although Theorem 2 suggests that we truncate the latent distribution, there are several limitations with this approach. First, the latent distribution cannot be truncated without knowing a closed form expression for $P_G$. Second, we may learn the disconnected set $Z$ by training a mixture distribution for $P_{\boldsymbol{z}}$ as is done in (Ben-Yosef & Weinshall, 2018; Gurumurthy et al., 2017). The problem with

this is that the geometric shape of $Z$ is restricted to be spherical or hyperellipsoidal. Third, before truncating the latent space, we need to train a generative network to proportionally learn the data distribution, which is impossible to confirm.

Given these limitations, we introduce the use of ensembles of generative networks in Section 4.1. This class of models addresses the issues above as follows. First, we will not need to have access to $P_G$ in any way before or after training. Second, knowing the geometric shape of $Z$ is no longer an issue because each network in the ensemble is trained on the connected set $\mathcal{X}_k$ instead of the disconnected whole $\mathcal{X}$. Finally, since the $k$-th network will only need to learn the distribution of $\mathcal{X}_k$, we reduce the complexity of the learned distribution and do not have to confirm that the distribution learned is proportionally correct.

# 4 ENSEMBLES OF GANS AND PARAMETER SHARING

We demonstrate how to train ensembles of GANs practically and relate ensembles to single GANs, cGANs, and GM-GANs. We focus on feedforward (Goodfellow et al., 2016) GANs in this section for concreteness; therefore, we study an ensemble of discriminators as well as generators.

## 4.1 ENSEMBLE OF GANS VS. A SINGLE GAN

Given an ensemble of GANs, we will write $G_k : \mathbb{R}^\ell \to \mathbb{R}^p$ as the $k$-th generator with parameters $\boldsymbol{\theta}_{G_k} \in \mathbb{R}^{|\boldsymbol{\theta}_G|}$ for $k \in \{1, \ldots, K\}$, where $K$ is the number of ensemble networks. We assume that each of the generators has the same architecture, hence $|\boldsymbol{\theta}_{G_i}| = |\boldsymbol{\theta}_{G_j}|$ for all $i, j$; thus we drop the subscript and write $|\boldsymbol{\theta}_G|$. Likewise, we write $D_k : \mathbb{R}^p \to [0, 1]$ for the $k$-th discriminator with parameters $\boldsymbol{\theta}_{D_k} \in \mathbb{R}^{|\boldsymbol{\theta}_D|}$ since the discriminators all have the same architecture. The latent distribution is the same for each ensemble network: $P_{\boldsymbol{z}}$. The generated distributions will be denoted $P_{G_k}$.

For concreteness, we assume that $K$ is the number of classes in the data; for MNIST, CIFAR-10, and ILSVRC2012, $K$ would be 10, 10, and 1000, respectively. If $K$ is unknown, then an unsupervised approach (Hoang et al., 2018; Khayatkhoei et al., 2018) can be used. Define the parameter $\boldsymbol{\pi} \in \mathbb{R}_+^K$ such that $\sum_{k=1}^K \pi_k = 1$. We then draw a one-hot vector $\boldsymbol{y} \sim \mathrm{Cat}(\boldsymbol{\pi})$ randomly and generate a sample using the $k$-th generator if the $k$-th component of $\boldsymbol{y}$ is 1. Hence, we have that a generated sample is given by $\boldsymbol{x} = G_k(\boldsymbol{z})$. This ensemble of GANs is trained by solving

$$\min_{\boldsymbol{\theta}_{G_k}} \max_{\boldsymbol{\theta}_{D_k}} V(\boldsymbol{\theta}_{G_k}, \boldsymbol{\theta}_{D_k}) \tag{2}$$

for $k \in \{1, \ldots, K\}$. Note that with an ensemble like this, the overall generated distribution $P_G(\boldsymbol{x}) = \sum_{k=1}^K \pi_k P_{G_k}(\boldsymbol{x})$ is a mixture of the ensemble distributions. This makes comparing a single GAN to an ensemble challenging; for example, consider comparing a Gaussian to a mixture of Gaussians.

In order to compare a single GAN to an ensemble of GANs, we define a new hybrid optimization

$$\min_{\boldsymbol{\theta}_{G_1}, \ldots, \boldsymbol{\theta}_{G_K}} \left( \max_{\boldsymbol{\theta}_{D_1}, \ldots, \boldsymbol{\theta}_{D_K}} \sum_{k=1}^K V(\boldsymbol{\theta}_{G_k}, \boldsymbol{\theta}_{D_k}) \text{ s.t. } \sum_{\substack{k=1 \\ j=k}}^K \|\boldsymbol{\theta}_{D_j} - \boldsymbol{\theta}_{D_k}\|_0 \leqslant t \right) \text{ s.t. } \sum_{\substack{k=1 \\ j=k}}^K \|\boldsymbol{\theta}_{G_j} - \boldsymbol{\theta}_{G_k}\|_0 \leqslant t, \tag{3}$$

where $\| \cdot \|_0 = 1$ denotes the $\ell_0$ "norm," which counts the number of non-zero values in a vector. Thus, $t \geqslant 0$ serves as a value indicating how many parameters are the same across different networks, which is more general than having tied weights between networks (Ghosh et al., 2018). We penalize the parameters because it is convenient, although it is not equivalent to, penalizing the functions themselves. This is true because $\boldsymbol{\theta}_{G_k} - \boldsymbol{\theta}_{G_j} = 0 \implies G_k - G_j = 0$ but the converse is not true. We analyze the behavior of (3) as we vary $t$ in the next theorem.

**Theorem 3.** *Let $G$ and $D$ be the generator and discriminator network in a GAN. Suppose that for $k \in \{1, \ldots, K\}$ we have that $G_k$ and $D_k$ have the same architectures as $G$ and $D$, respectively. Moreover, assume that $P_\mathcal{X}(\mathcal{X}_j) = P_\mathcal{X}(\mathcal{X}_k)$ for all $j, k$. Then,*

   i) *Suppose that $t \geqslant \max\left\{ K\frac{K-1}{2}|\boldsymbol{\theta}_D|, K\frac{K-1}{2}|\boldsymbol{\theta}_G| \right\}$. Then for all $k \in \{1, \ldots, K\}$ we have that $\left(\boldsymbol{\theta}_{G_k}^*, \boldsymbol{\theta}_{D_k}^*\right)$ is a solution to (3) if and only if $\left(\boldsymbol{\theta}_{G_k}^*, \boldsymbol{\theta}_{D_k}^*\right)$ is a solution to (2).*

ii) *Suppose that $t = 0$. Then we have that $(\boldsymbol{\theta}_G^*, \boldsymbol{\theta}_D^*)$ is a solution to (3) for each $k \in \{1, \ldots, K\}$ if and only if $(\boldsymbol{\theta}_G^*, \boldsymbol{\theta}_D^*)$ is a solution to (1).*

Informally, Theorem 3 shows that, when $t = 0$, we essentially have a single GAN, because all of the networks in the ensemble have the same parameters. If $t$ is large then we have an unconstrained problem such that the ensemble resembles the one in Equation (2). Therefore, this hybrid optimization problem trades off the parameter sharing between ensemble components in a way that allows us to compare performance of single GANs with ensembles.

Unfortunately, Equation (3) is a combinatorial optimization problem and is computationally intractable. Experimentally, we relax Equation (3) to the following

$$\min_{\boldsymbol{\theta}_{G_1}, \ldots, \boldsymbol{\theta}_{G_K}} \left( \max_{\boldsymbol{\theta}_{D_1}, \ldots, \boldsymbol{\theta}_{D_K}} \sum_{k=1}^{K} V(\boldsymbol{\theta}_{G_k}, \boldsymbol{\theta}_{D_k}) - \lambda \sum_{\substack{k=1 \\ j=k}}^{K} \|\boldsymbol{\theta}_{D_j} - \boldsymbol{\theta}_{D_k}\|_1 \right) + \lambda \sum_{\substack{k=1 \\ j=k}}^{K} \|\boldsymbol{\theta}_{G_j} - \boldsymbol{\theta}_{G_k}\|_1 \quad (4)$$

in order to promote parameter sharing and have an almost everywhere differentiable regularization term that we can backpropagate through while training. Although Equation (4) is a relaxation of Equation (3), we still have the same asymptotic behavior when we vary $\lambda$ as when we vary $t$ as shown in Appendix A.

## 4.2 Optimality of ensembles fo GANs

This next theorem shows that if we are able to learn each component's distribution, $P_{\mathcal{X}_k}$, then an ensemble can learn the whole data distribution $P_{\mathcal{X}}$.

**Theorem 4.** *Suppose that $G_k^*$ is the network that generates $\mathcal{X}_k$ for each $k \in \{1, \ldots, K\}$, i.e. $P_{G_k^*} = P_{\mathcal{X}_k}$. Under Definition 1, we can learn each $G_k^*$ by solving (2) with $V$ being the objective function in Equation (1).*

We know from (Goodfellow et al., 2014) that a globally optimal solution is achieved when the distribution of the generated images equals $P_{\mathcal{X}}$. Hence, this theorem has an important consequence: Training an ensemble of networks is optimal under our current data model.

It is important to note that the condition "$G_k$ is the network that generates $\mathcal{X}_k$" is necessary but not too strong because we may have a distribution that cannot be learned by a generative network or that our network does not have enough capacity to learn. We do not care about such cases however, because we are studying the behavior of generative networks under Definition 1.

## 4.3 Relation of ensembles of GANs to other GAN architectures

**Relation to cGANs.** We compare a cGAN to an ensemble of GANs. Recall from Section 2.2 that a cGAN has parameters $\boldsymbol{\theta}_G$ and $\boldsymbol{\theta}_D$ that do not change with different labels but there are matrices $\boldsymbol{B}_G$ and $\boldsymbol{B}_D$ that do depend on the labels. Specifically, they solve the optimization problem Theorem 3 with the additional constraint that the only parameters that can be different are the biases in the first layer. For other variants of cGANs a similar result applies.

**Theorem 5.** *A cGAN is equivalent to an ensemble of GANs with parameter sharing among all parameters except for the biases in the first layer. Moreover, the optimization in (3) can be modified so that it is equivalent to the cGAN optimization problem.*

**Relation to GM-GANs.** Another generative network that is related to ensembles is the GM-GAN. The first layer in GM-GANs transforms the latent distribution from isotropic Gaussian into a mixture of Gaussians. This new layer plays a similar role as the $\boldsymbol{B}_{.,k}$ in the cGAN comparison above, meaning that GM-GANs solve the optimization problem (3) with the additional constraint that the only parameters that can be different are the parameters in the first layer.

**Theorem 6.** *A GM-GAN is equivalent to an ensemble of GANs with parameter sharing among all parameters except for the first layer. Moreover, the optimization in (3) can be modified so that it is equivalent to the GM-GAN optimization problem.*

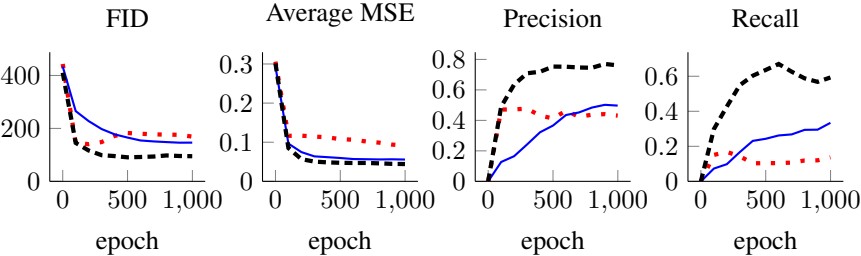

Figure 1: Ensembles of WGANs with fewer total parameters than a single WGAN perform better on CIFAR-10. We do not have to sacrifice computation to achieve better performance, we just need models that capture the underlying structure of the data. The **dotted red** line is the baseline WGAN, the **solid blue** line is the equivalent ensemble, and the **dashed black** line is the full ensemble.

## 5 EXPERIMENTAL RESULTS

In this section, we study how ensembles of WGAN (Arjovsky et al., 2017) compare with a single WGAN and a conditional WGAN. We use code from the authors' official repository (Arjovsky et al., 2018) to train the baseline model. We modified this code to implement our ensembles of GANs and cGAN. For evaluating performance, we use the FID score (Heusel et al., 2017; 2020), average MSE to the training data (Metz et al., 2016; Lipton & Tripathi, 2017), and precision/recall (Sajjadi et al., 2018; 2019). More details about the experimental setup are discussed in Appendix C.

### 5.1 ENSEMBLES PERFORM BETTER THAN SINGLE NETWORKS

We consider a basic ensemble of WGANs where we simply copy over the WGAN architecture 10 times and train each network on the corresponding class of CIFAR-10; we call this the "full ensemble". We compare this ensemble to the baseline WGAN trained on CIFAR-10.

Figure 1 shows that the full ensemble of WGANs performs better than the single WGAN. It is not immediately clear, however, whether this boost in performance is due to the functional difference of having an ensemble or if it is happening because the ensemble has more parameters. The ensemble has 10 times more parameters than the single WGAN, so the comparison is hard to make. Thus, we consider constraining the ensemble so that it has fewer parameters than the single WGAN.

### 5.2 ENSEMBLES WITH FEWER TOTAL PARAMETERS STILL OUTPERFORM A SINGLE NETWORK

The "equivalent ensemble" (3,120,040 total generator parameters) in Figure 1 still outperforms the single WGAN (3,476,704 generator parameters) showing that the performance increase comes from using the ensemble rather than just having larger capacity. In other words, considering ensembles of GANs allows for improved performance even though the ensemble is simpler than the original network in terms of number of parameters.

We see a performance boost as a result of increasing the number of parameters, in Figure 1. Therefore, we perform better because of having a better model (an ensemble) as well as by having more parameters. Now, we investigate a way that we can further improve performance.

### 5.3 PARAMETER SHARING AMONG ENSEMBLE COMPONENTS LEADS TO BETTER PERFORMANCE

We study how the regularization penalty $\lambda$ affects performance. As discussed in Section 4.1, we can learn a model that is somewhere between an ensemble and a single network by using $\ell^1$ regularization.

In Figure 2, the performance increases when we increase $\lambda$ in the equivalent ensemble from 0 to 0.001, implying that there is some benefit to regularization. Recall that by having $\lambda > 0$, we force parameter sharing between generator and discriminator networks. This performance increase is likely data dependent and has to do with the structure of the underlying data $\mathcal{X}$. For example, we can have pictures of badgers ($\mathcal{X}_1$) and zebras ($\mathcal{X}_2$) in our dataset and they are disconnected. However, the

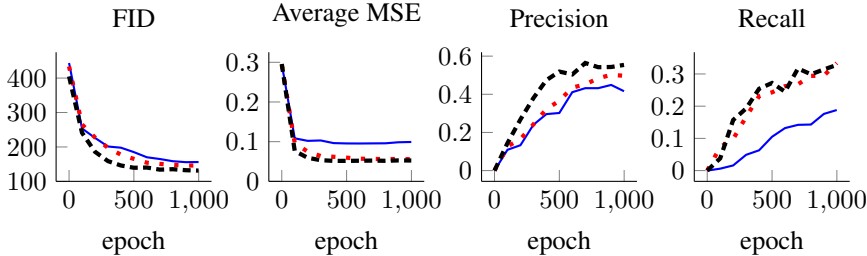

Figure 2: Ensembles of WGANs have a performance sweet spot when we regularize the optimization problem in expression (4) with different values of $\lambda$. Each curve is calculated using the equivalent ensemble of WGANs discussed in Section 5.2. We see that as we increase $\lambda$ to 0.001, the performance increases but then decreases when we continue to increase $\lambda$ to 0.01. This implies that there is an optimal value for $\lambda$ that can be found via hyperparameter tuning. The **solid blue** line is the equivalent ensemble with $\lambda = 0.01$, the **dotted red** line is the equivalent ensemble WGAN, and the **dashed black** line is the equivalent ensemble with $\lambda = 0.001$.

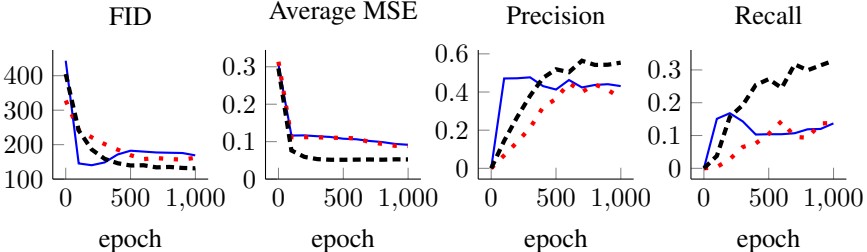

Figure 3: Regularized ensembles of WGANs using the optimization in (4) outperform cWGANs, even though cGANs are a type of ensemble. Here, cWGAN actually performs similarly to the baseline WGAN even though it takes into consideration class information. The **solid blue** line is the baseline, the **dotted red** line is the cWGAN, and the **dashed black** line is the equivalent ensemble with $\lambda = 0.001$.

backgrounds of these images are likely similar so that there is some benefit in $G_1$ and $G_2$ treating these images similarly, if only to remove the background.

As we increase $\lambda$ from 0.001 to 0.01 we notice that performance decreases. This means that there is a sweet spot and we may be able to find an optimal $0 < \lambda^* < 0.01$ via hyperparameter tuning. We know that the performance is not monotonic with respect to $\lambda$ because it decreases and then increases again; in other words, the performance has a minima that is not at $\lambda = 0$ or $\lambda \to \infty$. The optimization problem in expression (4) therefore can be used to find a better ensemble than the equivalent ensemble used in Section 5.2 which still has fewer parameters than the baseline WGAN.

### 5.4 ENSEMBLES OUTPERFORM CGANS

We modify a WGAN to be conditional and call it cWGAN. This cWGAN is trained on CIFAR-10, and we compare cWGAN to ensembles of WGANs. We do this because we showed in Section 4.3 that cGANs are an ensemble of GANs with a specific type of parameter sharing.

As can be seen from Figure 3, ensembles perform better than the cWGAN. The baseline WGAN model actually performs similarly to the cWGAN, which implies that the conditioning is not helping in this specific case. We hypothesize that our model ($\lambda = 0.001$) performs better because there are more parameters that are free in the optimization, instead of just the bias in the first layer. Thus, although cGANs are widely used, an ensemble with the regularization described in Section 4.1 can outperform them because the ensemble captures the disconnected structure of the data better.

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

# A  PROOFS

We first show that under Definition 1, our data is disconnected.

**Proof of Lemma 1.** Since $\overline{\mathcal{X}}_j \cap \overline{\mathcal{X}}_k = \mathcal{X}_j \cap \mathcal{X}_k = \varnothing$, we see that $\mathcal{X}$ is disconnected. $\qquad\square$

**Remark 1.** Note that if our data is disconnected, it doesn't necessarily follow Definition 1. This means that Definition 1 is a stronger condition than just having disconnected data. This can be seen by the following counter-example. We denote a truncated Gaussian as $\mathcal{N}(\mu, \sigma^2)|_S$, where the distribution is non-zero on $S$. Let

$$P_\mathcal{X} = \frac{1}{2}P_{\mathcal{X}_1} + \frac{1}{2}P_{\mathcal{X}_2} = \frac{1}{2}\mathcal{N}(0,1)|_{(-\infty,0)} + \frac{1}{2}\mathcal{N}(0,1)|_{(0,\infty)}$$

be the true distributions. Note that $(-\infty, 0)$ and $(0, \infty)$ are disconnected but do not follow Definition 1 because they are not compact. Moreover, we can learn this distribution easily by letting $G$ be the identity function and having $P_z = \mathcal{N}(0,1)$; it is trivial to show that this results in $P_G = P_\mathcal{X}$. Hence, disconnected data is too weak of an assumption—we need there to be a non-zero distance between our disconnected sets and that is what Definition 1 captures.

**Proof of Theorem 1.** Without loss of generality, we can assume that $G$ can generate at least from the two components $\mathcal{X}_1$ and $\mathcal{X}_2$. We define $B = G^{-1}(\mathcal{X})$ and note that $G(B)$ must be disconnected because $G$ can generate from at least $\mathcal{X}_1$ and $\mathcal{X}_2$ and they are disconnected. Since, $\mathcal{X}$ is disconnected and closed in $\mathbb{R}^p$, we have that $B$ is disconnected and closed in $\mathbb{R}^\ell$ because $G$ is continuous (Theorem 4.8 and 4.22 in (Rudin, 1964)). Since $B$ is closed in $\mathbb{R}^\ell$, this means that $\mathbb{R}^\ell \backslash B$ is an open set. Moreover, $\mathbb{R}^\ell \backslash B$ is not empty because we know that $\mathbb{R}^\ell$ is connected and $B$ is not. We also know that the Lebesgue measure $\lambda$ of a nonempty, open set is positive, hence we have that

$$\lambda(\mathbb{R}^\ell \backslash B) > 0.$$

Since $\lambda$ is equivalent to $P_z$, we have that $P_z(z \in \mathbb{R}^\ell \backslash B) > 0$. Thus,

$$P_G(\boldsymbol{x} \in \mathbb{R}^p \backslash \mathcal{X}) = P_z(\boldsymbol{z} \in G^{-1}(\mathbb{R}^p \backslash \mathcal{X})) = P_z(\boldsymbol{z} \in \mathbb{R}^\ell \backslash B) > 0,$$

as desired. $\qquad\square$

**Proof of Corollary 1.** Since the data lies only on $\mathcal{X}$, we know that $P_\mathcal{X}(\boldsymbol{x}_{\text{data}} \in \mathbb{R}\backslash\mathcal{X}) = 0$ for any valid probability measure. However, we have that $P_G(\boldsymbol{x} \in \mathbb{R}^p \backslash \mathcal{X}) > 0$. Hence, $d(P_G, P_\mathcal{X}) > 0$ for any metric $d$. $\qquad\square$

**Proof of Theorem 2.** The truncated latent distribution is denoted $P_{\boldsymbol{z}_T}$ and is defined as

$$P_{\boldsymbol{z}_T}(B) = \frac{P_z(B \cap Z)}{P_z(Z)}$$

for any set $B \in \mathcal{F}$. Hence, we have that

$$\begin{aligned}
P_{G|Z}(A) &= P_{\boldsymbol{z}_T}(G^{-1}(A)) \\
&= \frac{P_z(G^{-1}(A) \cap Z)}{P_z(Z)} \\
&= \frac{1}{P_z(Z)}P_z(G^{-1}(A) \cap G^{-1}(\mathcal{X})) \\
&= \frac{1}{P_z(Z)}P_z(G^{-1}(A \cap \mathcal{X})) \\
&= \frac{1}{P_z(Z)}P_G(A \cap \mathcal{X}) \\
&= \frac{c}{P_z(Z)}P_\mathcal{X}(A \cap \mathcal{X}) \\
&= \frac{c}{P_z(Z)}P_\mathcal{X}(A)
\end{aligned}$$

for any $A \in \mathcal{F}'$. The last equality is true because $P_{\mathcal{X}}(\mathcal{X}) = 1$, so that any points outside of $\mathcal{X}$ have zero probability. For the result above, set $A = \mathbb{R}^p$ to see that $c = P_{\boldsymbol{z}}(Z)$, implying that

$$P_{G|_z}(A) = P_{\mathcal{X}}(A) \qquad A \in \mathcal{F}',$$

as desired. $\qquad\qquad\qquad\qquad\qquad\qquad\qquad\qquad\qquad\qquad\qquad\qquad\qquad\qquad\qquad\quad$ $\square$

**Proof of Theorem 3.** First we prove i). If $t \geqslant \max\left\{ K^{\frac{K-1}{2}}|\boldsymbol{\theta}_D|, K^{\frac{K-1}{2}}|\boldsymbol{\theta}_G| \right\}$ then the constraints on (3) are unnecessary so that the problem reduces to

$$\min_{\boldsymbol{\theta}_{G_1},\dots,\boldsymbol{\theta}_{G_K}} \max_{\boldsymbol{\theta}_{D_1},\dots,\boldsymbol{\theta}_{D_K}} \sum_{k=1}^{K} V(\boldsymbol{\theta}_{G_k}, \boldsymbol{\theta}_{D_k}) = \min_{\boldsymbol{\theta}_{G_1},\dots,\boldsymbol{\theta}_{G_K}} \sum_{k=1}^{K} \max_{\boldsymbol{\theta}_{D_k}} V(\boldsymbol{\theta}_{G_k}, \boldsymbol{\theta}_{D_k})$$

$$= \sum_{k=1}^{K} \min_{\boldsymbol{\theta}_{G_k}} \max_{\boldsymbol{\theta}_{D_k}} V(\boldsymbol{\theta}_{G_k}, \boldsymbol{\theta}_{D_k}),$$

which is equivalent to solving the optimization problem

$$\min_{\boldsymbol{\theta}_{G_k}} \max_{\boldsymbol{\theta}_{D_k}} V(\boldsymbol{\theta}_{G_k}, \boldsymbol{\theta}_{D_k}), \quad k \in \{1, \dots, K\}.$$

Thus, i) is shown.

Next, we prove ii). Suppose that $t = 0$. Note that given a distribution $P_{\mathcal{X}}$, we can restrict it to each component $\mathcal{X}_k$ and normalize to get the restricted distributions

$$P_{\mathcal{X}_k}(A) = \frac{P_{\mathcal{X}}(A)}{P_{\mathcal{X}}(\mathcal{X}_k)}$$

for each $A \in \mathcal{F}'$ and each $k$. Since we assume that $P_{\mathcal{X}}(\mathcal{X}_j) = P_{\mathcal{X}}(\mathcal{X}_k)$ for all $j, k$ and that $P_{\mathcal{X}}(\mathcal{X}) = \sum_{k=1}^{K} P_{\mathcal{X}}(\mathcal{X}_k) = 1$, we see that $P_{\mathcal{X}}(\mathcal{X}_k) = \frac{1}{K}$ for each $k$. This implies that for any measurable function $f : \mathbb{R}^p \to \mathbb{R}$, we have that

$$\sum_{k=1}^{K} \mathbb{E}_{\boldsymbol{x} \sim P_{\mathcal{X}_k}}[f(\boldsymbol{x})] = \sum_{k=1}^{K} \int_{\boldsymbol{x} \in \mathcal{X}_k} f(\boldsymbol{x}) P_{\mathcal{X}_k}(d\boldsymbol{x})$$

$$= K \sum_{k=1}^{K} \int_{\boldsymbol{x} \in \mathcal{X}_k} f(\boldsymbol{x}) P_{\mathcal{X}}(d\boldsymbol{x})$$

$$= K \int_{\boldsymbol{x} \in \mathcal{X}} f(\boldsymbol{x}) P_{\mathcal{X}}(d\boldsymbol{x}) \qquad \left( \text{Since } \mathcal{X} = \bigcup_{k=1}^{K} \mathcal{X}_k \right)$$

$$= K \mathbb{E}_{\boldsymbol{x} \sim P_{\mathcal{X}}}[f(\boldsymbol{x})].$$

Suppose that V is the standard cross entropy objective function. We will use the notation $V(\boldsymbol{\theta}_G, \boldsymbol{\theta}_D; P)$ to show that we are evaluating $V(\boldsymbol{\theta}_G, \boldsymbol{\theta}_D)$ with the data distribution $P$. Then, we see that

$$\sum_{k=1}^{K} V(\boldsymbol{\theta}_G, \boldsymbol{\theta}_D; P_{\mathcal{X}_k}) = \sum_{k=1}^{K} \mathbb{E}_{\boldsymbol{x} \sim P_{\mathcal{X}_k}}[\log D(\boldsymbol{x})] + \mathbb{E}_{\boldsymbol{z} \sim P_{\boldsymbol{z}}}[\log(1 - D(G(\boldsymbol{z})))]$$

$$= K \mathbb{E}_{\boldsymbol{x} \sim P_{\mathcal{X}}}[\log D(\boldsymbol{x})] + K \mathbb{E}_{\boldsymbol{z} \sim P_{\boldsymbol{z}}}[\log(1 - D(G(\boldsymbol{z})))]$$

$$= K V(\boldsymbol{\theta}_G, \boldsymbol{\theta}_D; P_{\mathcal{X}}).$$

Similarly, if $V$ is the Wasserstein objective function, then

$$\sum_{k=1}^{K} V(\boldsymbol{\theta}_G, \boldsymbol{\theta}_D; P_{\mathcal{X}_k}) = \sum_{k=1}^{K} \mathbb{E}_{\boldsymbol{x} \sim P_{\mathcal{X}_k}}[D(\boldsymbol{x})] - \mathbb{E}_{\boldsymbol{z} \sim P_{\boldsymbol{z}}}[D(G(\boldsymbol{z}))]$$

$$= K \mathbb{E}_{\boldsymbol{x} \sim P_{\mathcal{X}}}[D(\boldsymbol{x})] - K \mathbb{E}_{\boldsymbol{z} \sim P_{\boldsymbol{z}}}[D(G(\boldsymbol{z}))]$$

$$= K V(\boldsymbol{\theta}_G, \boldsymbol{\theta}_D; P_{\mathcal{X}}).$$

This means that

$$\min_{\boldsymbol{\theta}_{G_1},...,\boldsymbol{\theta}_{G_K}} \left( \max_{\boldsymbol{\theta}_{D_1},...,\boldsymbol{\theta}_{D_K}} \sum_{k=1}^{K} V(\boldsymbol{\theta}_{G_k}, \boldsymbol{\theta}_{D_k}; P_{\mathcal{X}_k}) \text{ s.t. } \sum_{\substack{k=1 \\ j=k}}^{K} \|\boldsymbol{\theta}_{D_j} - \boldsymbol{\theta}_{D_k}\|_0 = 0 \right) \text{ s.t. } \sum_{\substack{k=1 \\ j=k}}^{K} \|\boldsymbol{\theta}_{G_j} - \boldsymbol{\theta}_{G_k}\|_0 = 0$$

$$= \min_{\boldsymbol{\theta}_G} \max_{\boldsymbol{\theta}_D} \sum_{k=1}^{K} V(\boldsymbol{\theta}_G, \boldsymbol{\theta}_D; P_{\mathcal{X}_k})$$

$$= \min_{\boldsymbol{\theta}_G} \max_{\boldsymbol{\theta}_D} KV(\boldsymbol{\theta}_G, \boldsymbol{\theta}_D; P_{\mathcal{X}}),$$

which is equivalent to the optimization problem in (1), as desired. $\square$

**Theorem 7.** *Let $G$ and $D$ be the generator and discriminator network in a GAN. Suppose that for $k \in \{1, \ldots, K\}$ we have that $G_k$ and $D_k$ have the same architectures as $G$ and $D$, respectively. Moreover, assume that $P_{\mathcal{X}}(\mathcal{X}_j) = P_{\mathcal{X}}(\mathcal{X}_k)$ for all $j, k$. Then,*

    *i) Suppose that $\lambda = 0$. Then for all $k \in \{1, \ldots, K\}$ we have that $(\boldsymbol{\theta}_{G_k}^*, \boldsymbol{\theta}_{D_k}^*)$ is a solution to (4) if and only if $(\boldsymbol{\theta}_{G_k}^*, \boldsymbol{\theta}_{D_k}^*)$ is a solution to (2).*

    *ii) Suppose that $\lambda \to \infty$. Then we have that $(\boldsymbol{\theta}_G^*, \boldsymbol{\theta}_D^*)$ is a solution to (4) for each $k \in \{1, \ldots, K\}$ if and only if $(\boldsymbol{\theta}_G^*, \boldsymbol{\theta}_D^*)$ is a solution to (1).*

**Proof of Theorem 7.** First we prove i). If $\lambda = 0$ then we have that

$$\lambda \sum_{\substack{k=1 \\ j=k}}^{K} \|\boldsymbol{\theta}_{D_j} - \boldsymbol{\theta}_{D_k}\|_1 = \lambda \sum_{\substack{k=1 \\ j=k}}^{K} \|\boldsymbol{\theta}_{G_j} - \boldsymbol{\theta}_{G_k}\|_1 = 0$$

on (4). Hence, the problem reduces to the unconstrained problem of (2).

Next, we prove ii). Since $\lambda \to \infty$, any solution where $\boldsymbol{\theta}_{D_k} \neq \boldsymbol{\theta}_{D_j}$ or $\boldsymbol{\theta}_{G_k} \neq \boldsymbol{\theta}_{G_j}$ for all $j, k \in \{1, \ldots, K\}$ is suboptimal. Consequently, it means that the optimization problem in (4) reduces to

$$\min_{\boldsymbol{\theta}_{G_1},...,\boldsymbol{\theta}_{G_K}} \left( \max_{\boldsymbol{\theta}_{D_1},...,\boldsymbol{\theta}_{D_K}} \sum_{k=1}^{K} V(\boldsymbol{\theta}_{G_k}, \boldsymbol{\theta}_{D_k}; P_{\mathcal{X}_k}) - \lambda \sum_{\substack{k=1 \\ j=k}}^{K} \|\boldsymbol{\theta}_{D_j} - \boldsymbol{\theta}_{D_k}\|_1 \right) + \lambda \sum_{\substack{k=1 \\ j=k}}^{K} \|\boldsymbol{\theta}_{G_j} - \boldsymbol{\theta}_{G_k}\|_1$$

$$= \min_{\boldsymbol{\theta}_G} \max_{\boldsymbol{\theta}_D} KV(\boldsymbol{\theta}_G, \boldsymbol{\theta}_D; P_{\mathcal{X}}),$$

which is equivalent to the optimization problem in (1). We mainly just outline the proof here because it is so similar to the proof of Theorem 3. $\square$

**Proof of Theorem 4.** Note that $P_{\mathcal{X}}$ is the total data distribution and that $P_{\mathcal{X}_k}$ is the distribution of each disconnected set. This means that

$$P_{\mathcal{X}} = \sum_{k=1}^{K} \pi_k P_{\mathcal{X}_k}$$

for some mixture coefficients $\alpha_k > 0$ so that $\sum_{k=1}^{K} \alpha_k = 1$.

Fix an arbitrary $k \in \{1, \ldots, K\}$. Since $P_{G_k^*} = P_{\mathcal{X}_k}$, we have that

$$\min_{\boldsymbol{\theta}_{G_k}} \max_{\boldsymbol{\theta}_{D_k}} V(\boldsymbol{\theta}_{G=k}, \boldsymbol{\theta}_{D_k}) = \min_{\boldsymbol{\theta}_{G_k}} \max_{\boldsymbol{\theta}_{D_k}} \mathbb{E}_{\boldsymbol{x} \sim P_{\mathcal{X}_k}}[\log D_k(\boldsymbol{x})] + \mathbb{E}_{\boldsymbol{z} \sim P_{\boldsymbol{z}}}[\log(1 - D_k(G_k(\boldsymbol{z})))]$$

has a solution of $P_{G_k^*} = P_{\mathcal{X}_k}$ from Theorem 1 of (Goodfellow et al., 2014). Since this is true for every $k$ and since $P_{\mathcal{X}} = \sum_{k=1}^{K} \pi_k P_{\mathcal{X}_k}$, we learn the complete data distribution. $\square$

**Proof of Theorem 5.** First we show that ensembles are equivalent to cGANs under the right conditions.

Fix the architecture of the networks considered and only focus on the generator. We want to show that cGANs and ensembles are equivalent, so we first focus on the generators in cGANs. We define the set of functions which represent conditional versions of the fixed architecture as

$$\mathcal{G}(K) = \left\{ G_{\boldsymbol{\theta},\boldsymbol{B}} : \mathbb{R}^{\ell} \times \{1, \ldots, K\} \to \mathbb{R}^p : \boldsymbol{B}, \boldsymbol{\theta} \text{ are network parameters} \right\},$$

where $\boldsymbol{B}$ is a matrix with $K$ columns and whose rows depend on the width of the first hidden layer as discussed in Section 2.1. The rest of the parameters are represented as the vector $\boldsymbol{\theta}$ above. It is clear that for a function in $\mathcal{G}(K)$, there could be many corresponding networks; some of these are due to activation and weight symmetries (Bishop, 2006). This implicitly create an equivalence class of networks; in particular, two networks are equivalent if they are the same mapping, regardless of weights. These symmetries do not affect our argument but it a subtlety to keep in mind.

Obviously not every ensemble is the same as the generator in a cGAN; however, we focus on a very specific type of ensemble. We define the set of all ensembles that have a variable bias in the first layer as

$$\mathcal{G}_E(K) = \left\{ \left( G_{\boldsymbol{\theta},\boldsymbol{b}_k} : \mathbb{R}^{\ell} \to \mathbb{R}^p \right)_{k=1}^{K} : \boldsymbol{b}_k, \boldsymbol{\theta} \text{ are network parameters for each } k \right\},$$

which is a collection of $K$-tuples of functions that represent networks with our fixed architecture. In the definition above, for $j \neq k$, we see that $G_{\boldsymbol{\theta},\boldsymbol{b}_j}$ and $G_{\boldsymbol{\theta},\boldsymbol{b}_k}$ share the same parameter $\boldsymbol{\theta}$, but the biases $\boldsymbol{b}_j$ and $\boldsymbol{b}_k$ may be different. Ensembles that are used in the construction of $\mathcal{G}_E(K)$ are essentially ensembles that have parameter sharing everywhere except in the bias term of the first layer; these biases are not constrained to be similar at all.

Since a network that induces a function in $\mathcal{G}(K)$ is the conditional version of the networks in the ensembles that are used to construct $\mathcal{G}_E(K)$, as described in Section 2.1, then parameter equality implies functional equality. In other words, for $G_{\boldsymbol{\theta},[\boldsymbol{b}_1,\ldots,\boldsymbol{b}_K]} \in \mathcal{G}(K)$ and $(G_{\boldsymbol{\theta},\boldsymbol{b}_k})_{k=1}^{K} \in \mathcal{G}_E(K)$ we have that $G_{\boldsymbol{\theta},[\boldsymbol{b}_1,\ldots,\boldsymbol{b}_K]}(\boldsymbol{z}, k) = G_{\boldsymbol{\theta},\boldsymbol{b}_k}(\boldsymbol{z})$ for all $\boldsymbol{z} \in \mathbb{R}^{\ell}, k \in \{1, \ldots, K\}$ and all parameters $\boldsymbol{\theta}, \boldsymbol{b}_k$.

We will now show that there exists a one-to-one correspondence between these two sets. Suppose that $T : \mathcal{G}_E(K) \to \mathcal{G}(K)$ is defined by

$$\left( T\left( (G_{\boldsymbol{\theta},\boldsymbol{b}_j})_{j=1}^{K} \right) \right)(\boldsymbol{z}, k) = G_{\boldsymbol{\theta},\boldsymbol{b}_k}(\boldsymbol{z}) = G_{\boldsymbol{\theta},[\boldsymbol{b}_1,\ldots,\boldsymbol{b}_K]}(\boldsymbol{z}, k)$$

for each $\boldsymbol{z} \in \mathbb{R}^{\ell}$ and $k \in \{1, \ldots, K\}$. Informally, we map an ensemble to a single network by just picking the $k$-th network in the ensemble. For a fixed $\boldsymbol{\theta}$ and $\boldsymbol{b}_1, \ldots, \boldsymbol{b}_K$ we see that $T\left( (G_{\boldsymbol{\theta},\boldsymbol{b}_j})_{j=1}^{K} \right)$ is indeed a function from $\mathbb{R}^{\ell} \times \{1, \ldots, K\}$ to $\mathbb{R}^p$. Moreover, $T\left( (G_{\boldsymbol{\theta},\boldsymbol{b}_j})_{j=1}^{K} \right)$ is equal to (as a function) to $G_{\boldsymbol{\theta},[\boldsymbol{b}_1,\ldots,\boldsymbol{b}_K]} \in \mathcal{G}(K)$ so that $T$ is well defined.

Let $G_{\boldsymbol{\theta},\boldsymbol{B}} \in \mathcal{G}(K)$ be given. Then we just let $\boldsymbol{b}_1, \ldots, \boldsymbol{b}_K$ be the columns of $\boldsymbol{B}$ and we see that $T\left( (G_{\boldsymbol{\theta},\boldsymbol{b}_j})_{j=1}^{K} \right) = G_{\boldsymbol{\theta},\boldsymbol{b}_k}$ implies that $T$ is surjective. Next suppose that $G_{\boldsymbol{\theta}^{\alpha},\boldsymbol{B}^{\alpha}} = G_{\boldsymbol{\theta}^{\beta},\boldsymbol{B}^{\beta}}$ are functions in $\mathcal{G}(K)$ with $\boldsymbol{B}^{\alpha} = [\boldsymbol{b}_1^{\alpha}, \ldots, \boldsymbol{b}_K^{\alpha}]$ and $\boldsymbol{B}^{\beta} = [\boldsymbol{b}_1^{\beta}, \ldots, \boldsymbol{b}_K^{\beta}]$. Then clearly we have that for $\left( G_{\boldsymbol{\theta}^{\alpha},\boldsymbol{b}_k^{\alpha}} \right)_{k=1}^{K}$ and $\left( G_{\boldsymbol{\theta}^{\beta},\boldsymbol{b}_k^{\beta}} \right)_{k=1}^{K}$ in $\mathcal{G}_E(K)$ that

$$G_{\boldsymbol{\theta}^{\alpha},\boldsymbol{b}_k^{\alpha}}(\boldsymbol{z}) = G_{\boldsymbol{\theta}^{\alpha},\boldsymbol{B}^{\alpha}}(\boldsymbol{z}, k) = G_{\boldsymbol{\theta}^{\beta},\boldsymbol{B}^{\beta}}(\boldsymbol{z}, k) = G_{\boldsymbol{\theta}^{\beta},\boldsymbol{b}_k^{\beta}}(\boldsymbol{z}),$$

implying that $T$ is injective.

Thus, $T$ is a one-to-one correspondence between $\mathcal{G}_E(K)$ and $\mathcal{G}(K)$. Hence, for every ensemble of networks defined above, we can find a cGAN which is equivalent to the ensemble. This equivalence is defined as the equivalence of the functions induced by these networks. The above result holds for any architecture and all $K \in \mathbb{Z}^{+}$. Thus, it also holds for the discriminator networks.

Next, we want to show that a modified version of the optimization problem from (3) yields the cGAN optimization problem.

We begin with the generic optimization problem from (3) and see that it can be rewritten as

$$\min_{\boldsymbol{\theta}_{G_1},\dots,\boldsymbol{\theta}_{G_K}} \left( \max_{\boldsymbol{\theta}_{D_1},\dots,\boldsymbol{\theta}_{D_K}} \sum_{k=1}^{K} V(\boldsymbol{\theta}_{G_k},\boldsymbol{\theta}_{D_k}) \text{ s.t. } \sum_{\substack{k=1\\j=k}}^{K} \|\boldsymbol{\theta}_{D_j} - \boldsymbol{\theta}_{D_k}\|_0 \leqslant t \right) \text{ s.t. } \sum_{\substack{k=1\\j=k}}^{K} \|\boldsymbol{\theta}_{G_j} - \boldsymbol{\theta}_{G_k}\|_0 \leqslant t$$

$$= \min_{\boldsymbol{\theta}_{G_1},\dots,\boldsymbol{\theta}_{G_K}} \left( \max_{\boldsymbol{\theta}_{D_1},\dots,\boldsymbol{\theta}_{D_K}} \sum_{k=1}^{K} V(\boldsymbol{\theta}_{G_k},\boldsymbol{\theta}_{D_k}) \text{ s.t. } C_D \right) \text{ s.t. } C_G$$

$$= \min_{\boldsymbol{\theta}_{G_1},\dots,\boldsymbol{\theta}_{G_K}} \left( \max_{\boldsymbol{\theta}_{D_1},\dots,\boldsymbol{\theta}_{D_K}} \sum_{k=1}^{K} V\left( \begin{bmatrix} \boldsymbol{\theta}'_{G_k} \\ (\boldsymbol{B}_G)_{\cdot,k} \end{bmatrix}, \begin{bmatrix} \boldsymbol{\theta}'_{D_k} \\ (\boldsymbol{B}_D)_{\cdot,k} \end{bmatrix} \right) \text{ s.t. } C_D \right) \text{ s.t. } C_G,$$

where we simply use the name $C_D$ for the constraint $\sum_{\substack{k=1\\j=k}}^{K} \|\boldsymbol{\theta}_{D_j} - \boldsymbol{\theta}_{D_k}\|_0 \leqslant t$ and similarly for $C_G$.
This is purely for notational convenience. Likewise, we simply denote $\boldsymbol{\theta}_{G_k}$ as $[(\boldsymbol{\theta}'_{G_k})^T (\boldsymbol{B}_G)^T_{\cdot,k}]^T$ and similarly for $\boldsymbol{\theta}_{D_k}$, for each $k$. Keep in mind that $\boldsymbol{B}_G$ and $\boldsymbol{B}_D$ are matrices such that the $k$-th column is the the bias of the first layer of the $k$-th network in the ensemble. So far, we have only introduced notational changes.

Consider what happens if we change the constraints to

$$C'_D = \sum_{\substack{k=1\\j=k}}^{K} \|\boldsymbol{\theta}'_{D_j} - \boldsymbol{\theta}'_{D_k}\|_0 = 0$$

$$C'_G = \sum_{\substack{k=1\\j=k}}^{K} \|\boldsymbol{\theta}'_{G_j} - \boldsymbol{\theta}'_{G_k}\|_0 = 0.$$

We have that $\boldsymbol{B}_G$ and $\boldsymbol{B}_D$ are unconstrained and that $\boldsymbol{\theta}'_{G_k}$ is forced to be equal to $\boldsymbol{\theta}'_{G_j}$ for all $k$ and $j$. Similarly $\boldsymbol{\theta}'_{D_k} = \boldsymbol{\theta}'_{D_j}$ for all $k$ and $j$. Hence, we can say that the optimization problem above with the new constraint is

$$\min_{\boldsymbol{\theta}_{G_1},\dots,\boldsymbol{\theta}_{G_K}} \left( \max_{\boldsymbol{\theta}_{D_1},\dots,\boldsymbol{\theta}_{D_K}} \sum_{k=1}^{K} V\left( \begin{bmatrix} \boldsymbol{\theta}'_{G_k} \\ (\boldsymbol{B}_G)_{\cdot,k} \end{bmatrix}, \begin{bmatrix} \boldsymbol{\theta}'_{D_k} \\ (\boldsymbol{B}_D)_{\cdot,k} \end{bmatrix} \right) \text{ s.t. } C'_D \right) \text{ s.t. } C'_G$$

$$= \min_{\boldsymbol{\theta}_{G_1},\dots,\boldsymbol{\theta}_{G_K}} \max_{\boldsymbol{\theta}_{D_1},\dots,\boldsymbol{\theta}_{D_K}} \sum_{k=1}^{K} V\left( \begin{bmatrix} \boldsymbol{\theta}_G \\ (\boldsymbol{B}_G)_{\cdot,k} \end{bmatrix}, \begin{bmatrix} \boldsymbol{\theta}_D \\ (\boldsymbol{B}_D)_{\cdot,k} \end{bmatrix} \right),$$

which is equivalent to the cGAN optimization problem. Here, we just define $\boldsymbol{\theta}_G$ to be shorthand for any one of the $\boldsymbol{\theta}_{G_k}$ vectors, since they are all the same.

Hence, a cGAN is equivalent to solving the ensemble optimization problem in (3) with a modified constraint.

$\square$

**Proof of Theorem 6.** The proof for this is very similar to the proof for Theorem 5. $\square$

## B    ESTIMATION OF ENSEMBLE PARAMETERS

In Section 4.1, we assume that $k \sim p_k$ is a multinomial distribution of degree $K$ parameters: $\pi_i$ for $i = 1,\dots,K$. Using the maximum likelihood estimator (Bishop, 2006) we obtain

$$\hat{\pi}_i^{\text{MLE}} = \frac{1}{N} \sum_{j=1}^{N} \mathbb{1}(\boldsymbol{y}_j = i)$$

for $i = \{1,\dots,K\}$. For datasets like MNIST (LeCun et al., 1998) and CIFAR-10 (Krizhevsky & Hinton, 2009), $k$ is a uniformly distributed random variable. For others one may have to calculate $p_k$ based on class imbalances.

## C    EXPERIMENTAL DETAILS

In this section we describe the details of our experiments.

### C.1    PERFORMANCE MEASURES

We use FID (Heusel et al., 2017), average MSE (Metz et al., 2016), precision, and recall (Sajjadi et al., 2018) to evaluate our models.

For FID, precision, and recall we use the official repositories (Heusel et al., 2020; Sajjadi et al., 2019). For each of these methods, we compare a set of generated images to a set of images from the training set. For the FID calculation, we use the precalculated statistics for CIFAR-10 and compare to 10,000 generated images from our trained networks. For precision and recall, we compare 10,000 generated images to 10,000 images in the training set. All other parameters are left the same.

For the average MSE calculation, we use the algorithm introduced in (Lipton & Tripathi, 2017), which was empirically shown to work 100% of the time on DCGAN architecture, such as WGAN. We modified the code in (Lao, 2017) so that it can be run with multiple restarts if desired. We ran our experiments with 1000 iterations and 5 restarts. We ran the code on 100 training images.

### C.2    BASELINE MODEL

For the baseline model, we ran the default WGAN code for 1000 epochs on CIFAR-10. All other parameters are left at their default values.

### C.3    FULL ENSEMBLE

To create the full ensemble, we just copied over the baseline model 10 times and trained each network pair $(G_k, D_k)$ in the ensemble on a single class of CIFAR-10. The training also lasted for 1000 epochs. This is equivalent to solving the optimization problem in (2).

### C.4    EQUIVALENT ENSEMBLE

Normally WGAN is trained with the following two architecture parameters: ngf = 64 and ndf = 64. However, to get 10% of the parameters we trained each ensemble component with ngf = 15 and ndf = 20. The depth of the generator and discriminator in the equivalent ensemble are the same as in the single WGAN, however, we modify the width of each corresponding layer so that the total parameters are fewer in the ensemble than in the single WGAN. Specifically, the generator of the WGAN has $3,576,704$ parameters and each generator of the equivalent ensemble has $312,004$ parameters. The discriminator of the WGAN has $2,765,568$ parameters and each discriminator of the equivalent ensemble has $272,880$ parameters. Reducing the width of each layer is not necessarily the optimal way to reducing parameters in a network. We do this because it is easy and effective, not because we are trying to reduce parameters in an optimal way, which is out of the scope of this paper. This is equivalent to solving the optimization problem in (2).

### C.5    REGULARIZED ENSEMBLES

For all the ensembles with $\lambda > 0$, we use the equivalent ensemble architecture, while solving (4).

### C.6    THE CGAN MODEL

For this architecture, we modify the baseline architecture and concatenate the class label, represented as a one-hot vector, to the input of the generator and discriminator networks.

