# OpenReview forum: "Ensembles of Generative Adversarial Networks for Disconnected Data"
_ICLR.cc/2021/Conference — Reject_

### Official Review · AnonReviewer1 · 2020-10-16
**Clear paper, interesting idea, underdeveloped exploration**

**Rating:** 4
**Confidence:** 3

**Review:**

Summary:

This work proposes that most relevant datasets to the machine learning community today have support on a mixture of disconnected components. They argue that popular GAN models cannot fit distributions of this kind and provide a number of proofs to convince the reader of this claim. The authors discuss a number of simple modifications on top of standard GANs which can alleviate this issue. These include replacing the standard unimodal latent distribution of the generator with a mixture model and replacing the standard generator with an ensemble of generators. The authors demonstrate that by replacing the standard GAN with an ensemble (or a pseudo-ensemble) they can achieve improved performance over a standard GAN given a fixed parameter budget with respect to a number of different evaluation metrics.


Strong areas:

I found this paper very clear and easy to follow which I very much appreciate.  I feel the theorems were motivated well and the correct amount of detail was shown in the main paper. To the best of my knowledge, all presented theory appears correct, but I am not an expert in that area. I think this paper asks the right kind of questions about the types of data we deal with. Beyond the “manifold hypothesis” there is not much discussion (to my knowledge) of other generic and reasonable assumptions which can be made about the types of data we see in the world. I think it is important to study how those properties of data interact with the properties of our models. Curitally, I think it is important we study how these properties positively reinforce each other (such as the manifold hypothesis and the low-dimensional latent spaces of a GAN) or how they negatively reinforce each other (like discussed in this work). So this paper focuses on what I believe to be an important set of issues.

Weaknesses:

I think it is widely accepted that most of the models used today cannot actually fit the distributions of the data we train them on. There are many ways in which the models we use can be deficient and this paper presents one such way which few people are likely to disagree with. Beyond this observation, there are many, equally valid, reasons why our models may be deficient. For example, the data may lie on a lower dimensional manifold which has a different topological structure than the base distribution of our generator (such as a ring). This is an equally obvious concern and similar theory to that presented in this work could be used to demonstrate this. But, the solutions presented here will not fix that issue. If the authors believe that the disconnected support issue is significantly more important than other topological issues then they should discuss this and provide evidence for why the disconnectivity issue is more important. The disconnectivity issue is a subset of the general topology issue so further evidence is needed to motivate study on this specific issue ignoring the larger problem.

Beyond this, I am left wondering why the authors restricted themselves to studying GANs. It seems to me that alternative classes of generative models will suffer from the same problems such as normalizing flows and VAEs (which are almost identical to GAN generators). It also seems to me that these models might be better candidates to study this effect due to their ability to compute exact likelihoods (or likelihood estimates in the case of VAEs).

The theory provided in this work was focused on what kind of distributions can be modeled by passing a fixed distribution through a learned, nonlinear mapping. For this reason, I feel the focus on GANs is very limiting since many popular models work like this. I think the paper would be much more beneficial to the community if they addressed these concerns in other classes of model.

Some more nit-picky issues:

Figures 1,2,3 would be more clear with a legend. I found myself going back and forth to the caption to figure out what was going on there. I am also concerned by the reported scores in these figures. It seems like the FIDs reported here are >100 which seems much higher than simple GANs (WGAN-GP gets around 36 I believe). Beyond that, I do not see much value in the way these results are presented. The plots allow us to see how these quantities evolve over training but that is not really discussed in the paper. I feel like a table would be more useful since some of the values do appear quite close.

My recommendation:

For the reasons stated above, I would recommend to reject this paper. From my perspective, the main focus of the paper feels too limited to have much impact. It seems to me there are a number of related issues that are just as pernicious which are not addressed at all in this work. If the authors provided more discussion of these issues and provided a convincing explanation as to why they are not as important as the disconnected sets issue, then I would be inclined to review this paper more favorably. Along those lines, I feel the theory developed here could be useful to study many classes of generative models. If the authors applied their theory to these additional models, I feel like this could be a much stronger contribution. This type of problem has been addressed within the flow community (one such reference is https://arxiv.org/pdf/2002.07101.pdf but there are many others along this line). The approach presented here is different and would be interesting to compare with the solutions provided by the flow community.

---

### Official Review · AnonReviewer3 · 2020-10-27
**Review for "Ensembles of Generative Adversarial Networks for Disconnected Data"**

**Rating:** 5
**Confidence:** 3

**Review:**

This paper shows that ensembles of GANs are better at learning disconnected distributions than a single GAN. They introduce a optimization procedure that allows to optimize the degree of parameter sharing between different GANs models in an ensemble. The theoretical findings are supportede empirically by experiments on the CIFAR-10 dataset.

While the problem setting is an important one I fail to see the main novelty in this paper. The fact that continuous functions keep the connectedness of the input space in tact is not a novel finding and explains why a single GAN with an input vector sampled from a Gaussian (or similar) cannot model a disconnected data distribution.

In this respect, I believe Section 4.1 and especially equations three and four are the more interesting part of the paper. While other papers have already showed that training ensembles of GANs can increase the performance, eq (4) offers a way to control the parameter sharing in GAN ensembles which might help with regularization.However, the experiments for the approach outlined in section 4.1 are very sparse. Only one GAN (relatively "old") architecture is evaluated (WGAN) and on only one relatively small dataset (CIFAR-10).

Since this approach introduces a new hyperparameter to control the effect of parameter-sharing more experiments on more datasets and GAN architectures are needed to verify the effectiveness. It would also be helpful to have the numeric results of Figures 1-3 instead of only graphs. E.g. in Fig 1 it often looks like the baseline WGAN and the equivalent ensemble (i.e. similar amount of parameters) have similar performance. All Figures show that the ensemble methods perform much better in the recall metric and it would be interesting to evaluate if this means that ensemble methods also improve mode dropping behavior and also how more modern GANs perform here.

Overall:
- I think equations (3) and (4) are the main novelty in this paper and the experiments should be expanded to evaluate this more
- for this I suggest to perform experiments with more modern architectures (at least WGAN-GP) and on more complex datasets (CIFAR 100, ImageNet, subsets of LSUN, etc).
- the experiments should then evaluate how much improvement we can actually achieve with ensembles and how different values of lambda affect the outcome on different GAN models and datasets

Minor: Section 4.2 has a spelling mistake in its title (should be "for" instead of "fo" I guess).

Some missing related work:
Kundu, J. N., Gor, M., Agrawal, D., & Babu, R. V. (2019). Gan-tree: An incrementally learned hierarchical generative framework for multi-modal data distributions. In Proceedings of the IEEE International Conference on Computer Vision (pp. 8191-8200).

Tanielian, U., Issenhuth, T., Dohmatob, E., & Mary, J. (2020). Learning disconnected manifolds: a no GANs land. International Conference on Machine Learning (pp. 6767-6776).

---

### Official Review · AnonReviewer4 · 2020-10-30
**Interesting theoretical and experimental results on the effectiveness of ensemble of GANs for modeling disconnected data.**

**Rating:** 7
**Confidence:** 4

**Review:**

**Summary of contributions:**
This paper proposes to study how to model distributions with disconnected support. They show that a single continuous generator cannot model such distributions correctly. However they show that using an ensemble of generator instead of a single generator can indeed model such distributions. They then show that training an ensemble of GANs can be formulated as a constraint optimization problem, they show that this general formulation encompass several other GAN variants by varying the constraint, such as the standard GAN formulation, conditional GAN and gaussian-mixture GAN. Finally they show experimentally that ensemble GANs outperforms other GANs formulation on CIFAR.

**Pros:**
I find the paper quite well written and clear. The ideas are interesting and convincing, they rigorously show theoretically and experimentally the advantage of using ensemble methods for training GANs. I believe this work is significant and could lead to breakthrough in generative modeling or other settings where we might need to model distributions with disconnected support.

**Cons:**
- The theory seems to suggest that we only need to have an ensemble of generator and that we might not need to have an ensemble of discriminator. Might be interesting to see what happens if we use a single discriminator ? do we observe a big drop of performance ? If the theory is correct the performance shouldn't drop too much.
- The experiments are quite convincing, I think they could be even more convincing with a toy experiment where the distribution clearly has disconnected support. It would also be nice to have a comparison to GM-GAN. The CIFAR dataset is also quite limited and larger scale experiment such as ILSVRC would be very interesting to see, but I understand that this requires a lot of computational ressources and can be hard to do. Finally it's always nice to have some confidence interval for the empirical results.

**Minor comments or Typos:**
- Typo in theorem 1: $i \in \{2, ..., K\}$
- In theorem 2: State explicitly that the condition needs to be valid for $\forall A \in \mathcal{F}', A \subset \mathcal{X}$.
- In section 3.3 about truncated distribution, the argument about having a mixture distribution for the prior on the latent Z being problematic is not very clear. While this indeed restrict the shape of the prior, the generative network should be able to compensate this, thus as long as the support of Z is disconnected the generator should be able to learn distribution with disconnected support.
- In theorem 5 it would be nice to explicitly state how the optimization problem (3) is modified.

---

### Official Review · AnonReviewer2 · 2020-11-02
**Interesting analysis but fairly obvious results**

**Rating:** 4
**Confidence:** 4

**Review:**

This paper addresses the problem that models like GANS which learn continuous mappings from a connected latent space, are incapable of producing a mapping that contains disconnected components. The authors argue that real-world classes like "badger" and "zebra" are indeed disconnected and thus the inability of GANs to learn a disconnected mapping poses a problem. The authors formalize the statement and argue instead that one should learn an ensemble of GANs. These aren't really ensembles as in classification but components of a mixture model. They are using each GAN to represent one component of a mixture distribution. They also argue that conditional GANs and GANs with mixture distributions over the latents are similar. The paper closes with interesting but insufficient experiments showing the benfits of ensembles of GANs on one dataset (CIFAR10).

The paper is clear and nice to read. However the main arguments are actually fairly obvious and well known to all theorists that have worked on these problems. In fact I know multiple people that have discussed these points years ago, not as a paper contribution in themselves but as a jumping off point. The first argument flows straightforwardly from basic facts about continuity and push-forward distributions.

One key issue is that the result in Theorem 1 is not so striking as it may appear. All it says is that there will be nonzero probability assigned to points outside of either of the data components. But how large must this probability be? It's easy to see with a simple 1D example on the number line that we might be able to make that probability vanishingly small, at the expense of ratcheting up the  Lipschitzness of the neural network to arbitrary extremes.  So disconnected distributions may still be learnable (in a learning theoretic sense) even if they are not exactly representable, with an error that could still approach 0, even if for any particular network the error would be nonzero.

The most novel result, and the only one that isn't already plainly evident is theorem 3, concerning a proposed objective for learning ensembles of GANs and its equivalence to the original GAN objective when certain conditions on the number of parameters that can differ between GANs is capped. Unfortunately, the paper does little to unpack intuition for why this result is true or why it is interesting.

Overall it's a great start and I think the authors are on the right path. The writing is clear and the problem is important. But the authors need to go deeper, clarify what precisely is the learnability problem, why their method is justified, and/or to contribute far more by way of experimental support for the proposed ensembling techniques.

---

### Decision · Program_Chairs · 2021-01-07
**Final Decision**

**Decision:**

Reject

**Comment:**

I am recommending rejection for this paper for the following reasons:

I agree that the main claim is an obvious consequence of the structure of the GAN generator and the prior.
I'm also not sure why the authors restricted their analysis to GANs, but that's not a super important point to me.
More important is that the experimental validation of the ensembling idea is way below the bar for this conference.
Moreover, I know the authors touched on this a bit, but all of these modern GAN variants that work on imagenet
are implicitly ensembling anyway through e.g. the conditioning input and the structure of the special batch-norm.
Finally, the authors didn't respond to the reviews.